# Novel Inhibitors of 2′-*O*-Methyltransferase of the SARS-CoV-2 Coronavirus

**DOI:** 10.3390/molecules27092721

**Published:** 2022-04-23

**Authors:** Alexey Sulimov, Danil Kutov, Ivan Ilin, Yibei Xiao, Sheng Jiang, Vladimir Sulimov

**Affiliations:** 1Research Computing Center, Lomonosov Moscow State University, 119992 Moscow, Russia; sulimovv@mail.ru (A.S.); ivan.ilyin@srcc.msu.ru (I.I.); 2School of Pharmacy, China Pharmaceutical University, Nanjing 211198, China; yibei.xiao@cpu.edu.cn (Y.X.); jiangsh9@gmail.com (S.J.)

**Keywords:** SARS-CoV-2, nsp16 inhibitor, docking, quantum chemistry, virtual screening

## Abstract

The COVID-19 pandemic is still affecting many people worldwide and causing a heavy burden to global health. To eliminate the disease, SARS-CoV-2, the virus responsible for the pandemic, can be targeted in several ways. One of them is to inhibit the 2′-*O*-methyltransferase (nsp16) enzyme that is crucial for effective translation of viral RNA and virus replication. For methylation of substrates, nsp16 utilizes *S*-adenosyl methionine (SAM). Binding of a small molecule in the protein site where SAM binds can disrupt the synthesis of viral proteins and, as a result, the replication of the virus. Here, we performed high-throughput docking into the SAM-binding site of nsp16 for almost 40 thousand structures, prepared for compounds from three libraries: Enamine Coronavirus Library, Enamine Nucleoside Mimetics Library, and Chemdiv Nucleoside Analogue Library. For the top scoring ligands, semi-empirical quantum-chemical calculations were performed, to better estimate protein–ligand binding enthalpy. Relying upon the calculated binding energies and predicted docking poses, we selected 21 compounds for experimental testing.

## 1. Introduction

Pandemic coronavirus disease 2019 (COVID-19) is one of the most serious challenges we face [1]. The struggle against this disease requires the solving of various problems, including the urgent development of antiviral drugs with direct action on the therapeutic target proteins of severe acute respiratory syndrome coronavirus 2 (SARS-CoV-2) that caused the pandemic [2]. By the beginning of the pandemic, deep knowledge of the mechanisms of coronavirus replication at the molecular level had been gained thanks to studies of earlier diseases such as SARS [3] and MERS [4]. Currently, several therapeutic target proteins of this coronavirus are well established [5,6,7,8,9]. In this work, we focus our study on the SARS-CoV-2 2′-*O*-methyltransferase (2′-*O*-MTase; EC 2.1.1.296), which is referred to henceforth as nsp16 [10]. Non-Structural Protein 16 (nsp16), a viral RNA 2′-*O*-MTase, is one of the highly viable targets for the design of agents against SARS-CoV-2. Nsp16, with its cofactor nsp10, forms a heterodimer and stimulates 2′-*O*-methyltransferase activity. 2’-*O*-methylation is essential for formation of the cap structure in viral RNAs. The presence of the cap part on RNA of SARS-CoV-2 allows it to make its genetic information readable to the host translation machinery. It has been shown that this 2′O-MTase is indispensable for replication of coronaviruses in cell cultures [11].

Given the importance of the RNA capping process for mRNA stability and translation, and as an immune evasion mechanism, RNA-capping machineries are an attractive target for antiviral-drug design [12,13]. In the viral RNA capping process, non-structural protein 16 (nsp16), an *S*-adenosyl methionine (SAM)-dependent 2′-*O*-MTase, methylates the first transcribed nucleotide at the ribose 2′-OH position, to form a cap-1 structure. To perform a methyl transfer, nsp16 utilizes SAM that contains an activated methyl group and represents a common co-substrate for methyl transfer reactions in living organisms. Blockage of the SAM-binding site of nsp16 results in inhibition of the methylation process catalyzed by this enzyme. 2′-*O*-methylation is essential for formation of the cap structure in coronavirus (CoV) RNAs, and, therefore, plays an important role in the replication of CoVs. Furthermore, 2′-*O*-methylation allows CoVs to escape the recognition of immune sensors and resist the type I interferon-mediated anti-infective immune response. Importantly, 2′-*O*-MTase/nsp16 is highly conserved in structure and function among various CoVs, together with its indispensable role in CoV replication and immune escape, making it an attractive and effective target for broad-spectrum antivirals.

After the first SARS outbreak, the search for nsp16 inhibitors of SARS-CoV-1 led to identification of sinefungin (**1**) (see Figure 1) as a potent suppressor of the methyltransferase activity of nsp16 [14]. Sinefungin is similar to *S*-adenosylmethionine, an endogenous cofactor of nsp16, and binds to the same pocket. Recently, the ability of sinefungin to block the methyltransferase activity of nsp16 in SARS-CoV-2 was confirmed by X-ray crystallography [11].

In the study by [15], several nsp16 inhibitors were designed using bioisosteric replacements of the sulfonium and amino acid substructures of the *S*-adenosylmethionine. Virtual close analogs of *S*-adenosylmethionine were docked into active site of nsp16. Activity testing of the synthesized prioritized compounds showed nanomolar to sub-micromolar IC_50_ values for five compounds. The best IC_50_ was 8 nM for methylbenzoic acid derivative (**2**). However, selectivity testing on human glycine *N*-methyltransferase revealed that all inhibitors are non-selective and also inhibit this methyltransferase. The main reason for the non-selectivity can be attributed to the very high similarity between the found nsp16 inhibitors and *S*-adenosylmethionine, a co-factor used by most methyltransferases.

Since nsp14 is another viral methyltransferase with a similar active site architecture, some scientific groups initiated a combined search for both nsp16 and nsp14 inhibitors. In the study of [16], docking-based high-throughput virtual screening of 7 million commercially-available drug-like compounds and *S*-adenosylmethionine analogs was performed. As a result, 80 virtual screening hits (39 against nsp14 and 41 against nsp16) were identified, which were purchased and tested using an enzymatic homogeneous time-resolved fluorescent energy transfer assay. Of these, nine compounds showed a micromolar inhibition activity with an IC_50_ less than 200 μM. The best inhibitor (**3**) was based on a sulfanyl-thiazole core and possessed an IC_50_ against nsp16 of 51 µM. Similarly to the above-mentioned study, most of the compounds showed poor selectivity against human glycine *N*-methyltransferase.

Pharmacophore-based screening for 48 million drug-like compounds of the Zinc database was used to identify nsp16 inhibitors in [17]. A 3D pharmacophore model was constructed based on the complex of nsp16 with sinefungin (PDB ID: 6WKQ). The 24 best-scoring ligands were then docked in the SAM-binding pocket. Finally, molecular dynamics (MD) simulation experiments for the three best compounds were carried out, as a refinement step. These simulations revealed one compound (**4**, Figure 2) with a triazine scaffold as the potential nsp16 inhibitor. Testing of its activity has not yet been performed.

In the study by [10], the authors applied an virtual drug repurposing approach to identify nsp16 inhibitors binding to active site of the enzyme. Shape-based screening among FDA approved drugs, followed by molecular dynamics (MD) simulation, revealed that raltegravir and maraviroc can bind tightly to the active site of the protein, compared to sinefungin, and could be potential candidates for inhibition of nsp16. No in vitro confirmation of their activity has been completed to date.

Another strategy to target nsp16, is disrupting the nsp16–nsp10 complex, without affecting the *S*-adenosylmethionine-binding site. Such an approach was used, for example, in [18]. The authors virtually screened the North African Natural Products database for compounds that can interact with the nsp10 interface and disturb the nsp10–nsp16 complex formation. They identified four compounds (genkwanin-6-*C*-beta-glucopyranoside, paraliane diterpene, 4,5-di-*p*-*trans*-coumaroylquinic acid, and citrinamide A—Figure 2) that showed the best binding affinity, predicted by AutoDock Vina. Of the four compounds, genkwanin-6-*C*-beta-glucopyranoside showed the most stable complex with nsp10 in a molecular dynamics simulation, but it was not tested in vitro.

In this study, virtual screening of the databases of on-the-shelf low molecular weight compounds using docking was carried out, followed by quantum-chemical calculations of the protein–ligand binding enthalpy, with the goal of finding inhibitors of nsp16. The inhibition strategy was to target the SAM-binding site of the enzyme. As a result, twenty one compounds were selected for experimental testing of their inhibition of the nsp16 target protein.

## 2. Results and Discussion

### 2.1. Nsp16 Structure Preparation

To create a model of nsp16, we used corresponding structures from the Protein Data Bank [19]. We found eight crystal structures of nsp16 co-crystallized with small molecules in the Protein Data Bank. Of these, four contain *S*-adenosylmethionine (SAM), an endogenous cofactor of nsp16, which carries an activated methyl group (see Figure 3). Two other complexes have crystallized sinefungin. Sinefungin is an unselective pan-inhibitor of methyltransferases, an analog of SAM. One complex contains SAM, together with RNA substrate (m7GpppA), and another is nsp16 with SAH, a conjugate form of SAM without a methyl group. The PDB IDs and resolution values for these complexes are listed in Table 1.

The SAM molecule was not parameterized in the MMFF94 force field [20], which was implemented in our SOL docking program [21,22]. Thus, structures with SAM were not taken into account when choosing complexes for further model preparation. To estimate the general flexibility of the active site of nsp16 and possible induced fit effects, we performed the alignment of three complexes: 6WKS, 6WKQ, and 6YZ1 (see Figure 4). No large differences were found after alignment, which could be related to the high similarity between the ligands co-crystallized with complexes under study.

For further model preparation, we chose two complexes with sinefungin, since it is the only inhibitor co-crystallized with nsp16. Both complexes possess a good resolution (6WKQ has the better one) and no large gaps in their structure (Table 2).

We carried out docking of the native ligand for both complexes. “Native ligand” means a ligand that is co-crystallized with its target protein and deposited with it in one PDB file. Native ligand docking allows one to perform primary estimation of the prepared model, to reproduce the binding of the known ligand, as well as to estimate the suitability of the chosen docking tool to model the interactions crucial for binding of the particular ligand to the given target. To perform native ligand docking, a ligand is extracted into a separate file from the initial PDB file and protonated at pH 7.4. The peculiarity of sinefungin after protonation is the presence of three charged atoms (see Figure 5).

In a crystal complex of sinefungin and nsp16, one of three groups ionized at pH = 7.4 does not have any interactions with the protein and is exposed to the solvent (see Figure 6). This can influence the positioning accuracy, since in most physics-based force fields (including MMFF94), the electrostatic term dominates in the primary scoring of poses, due to insufficient modelling of solvent-based electric-field screening.

Structures of nsp16 obtained from the PDB files were protonated at pH = 7.4 with the Aplite program [21,25]. The missing residues mentioned in Table 2 were not repaired, since they are far from the SAM-binding site of nsp16 and do not influence the accuracy of the docking procedure.

The docking of native ligands into the corresponding 6YZ1 and 6WKQ failed, in terms of RMSD from the initial ligand coordinates (see Table 3).

In Table 3, we can see that the RMSD between the positions of the crystallized and docked native ligand is too large for the protein structures from the 6YZ1 and 6WKQ PDB complexes. We assumed that this failure could be related to the above-mentioned fact that one of two charged amino groups is not implicated in any interactions that can bias the results of modeling, where the electrostatic forces dominate. This phenomenon is illustrated in Figure 7: the best energy docking pose in the active site of nsp16 after native ligand docking for 6YZ1 has all charged groups implicated in polar interactions with charged residues.

To remove the bias related to the ionic interaction dominance, we neutralized one charged amino group of sinefungin (the one that was solvent-exposed in the crystal complex, see Figure 5) and re-docked this neutralized form of sinefungin into the nsp16 models prepared on the basis of 6YZ1 and 6WKQ. As can be seen from Table 4, 6YZ1 failed again; whereas, native ligand docking for 6WKQ, after partial neutralization of sinefungin, was successful in terms of the RMSD value and the score of the SOL program.

In Table 4, two results of native docking for the protein model prepared using the 6WKQ PDB structure are presented. First, the results of docking of the native ligand with default genetic algorithm (GA) parameters (see Section 3.2) are presented in the 6WKQ row. Here, RMSD is low, the SOL score is sufficiently negative, but the population of the first cluster is equal to 1. This means that the reliability of this docking, i.e., the reliability of the global energy optimization, is low. To increase our confidence that we had, indeed, found the global energy minimum of the protein–ligand complex, we performed docking of the native ligand at elevated GA parameters, and the results are presented in Table 4 in the row of 6WKQ*. For this docking, we employed the parallel version of the SOL docking program, and the following GA parameters were used: population size was equal to 6,000,000, the number of generations was equal to 1500, and the number of GA independent runs was equal to 99. This docking took five hours to complete using 512 cores on the Lomonosov-2 supercomputer [26]. For the elevated GA parameters, the docking found practically the same ligand position corresponding to the global energy minimum, and the population of the first cluster increased to eight. This result supported the assumption that this ligand position, found either with the default or elevated GA parameters, corresponded to the global energy minimum.

To check the model based on 6WKQ, we, additionally, performed cross-docking for *S*-adenosylmethionine (SAH) from the 6WJT complex and preliminary aligned it with the protein structure from 6WKQ. The results of this docking can be considered satisfactory, and they are outlined in Table 5.

For the default GA parameters, the docking was not very good: RMSD > 2 Å, the population of the first cluster was only 1, and the SOL score was noticeably less negative than that of the native ligand from the 6WKQ complex. For the elevated GA parameters: population size = 6,000,000, number of generations = 1500, and number of GA runs = 99, the docking was successful: RMSD < 1 Å, the population of the first cluster was noticeably greater than 1, and the SOL score was even more negative than that of the native ligand. This docking took 3 h and 43 min, using 512 cores on the Lomonosov-2 supercomputer. The results presented in Table 5 show that a better accuracy in the positioning of the ligand can lead to noticeably better (more negative) SOL score values. In addition, one can see the importance of the population of the first cluster: a population equal to 1 can indicate failure of docking.

The need to use elevated GA parameters, both for native and cross-docking, can be attributed to the complex nature of SFG/SAH, not the protein system. Both organic molecules are zwitterions and possess seven rotational degrees of freedom, complicating conformational search during docking. Therefore, virtual screening of selected libraries was done with the default parameters.

### 2.2. Results of the Virtual Screening

A two-step virtual screening of the three databases of low molecular weight ligands using docking, followed by quantum-chemical calculations of the protein–ligand binding enthalpy, resulted in the selection of the 21 best candidates to become inhibitors of nsp16 for experimental testing. Here, a low molecular weight compound means an organic molecule with molecular weight less than 1000 Da. The results of calculations for the best ligands and their structures are presented in Table 6 and Figure 8 (ligands selected from the Enamine Coronavirus Library) and Figure 9 (ligands selected from the Chemdiv Nucleoside Analogue Library), respectively.

As can be seen from Figure 8 and Figure 9, most of top compounds from both libraries contain basic nitrogen, charged positively at pH 7.4, which is consistent with the chemical nature of SAM and the reported basic inhibitors of nsp16. The presence of a charged group helped select compounds form a salt bridge interaction with Asp131 and other negatively charged residues of the SAM-binding site. All compounds listed in Table 6 completely block the SAM-binding site of nsp16, according to the docking studies. Some of the top compounds from the Enamine Coronavirus Library contain fragments typical for nucleoside analogs: indazole (**Z2003934806**, **Z2394358664**) and pyrrolopyrimidine (**Z2968942047**). These scaffolds are privileged fragments for inhibiting, for example, kinases [27], enzymes which use adenosine triphosphate (nucleoside) for phosphate transfer, and, thereby, have the potential to bind to other enzymes that bind to nucleoside-like molecules, including nsp16. Similarly to sinefungin and SAM, three of the four top compounds from the Chemdiv Nucleoside Analogue Library belong to purines. The compound with the most negative SOL score (**Z2045761676**) belongs to the 4-aminopyridines. The best potential binder in terms of enthalpy calculated by PM7, **Z1631839283,** is a derivative of diamino-indan, with two aliphatic amino groups.

As a representative of nucleoside-like potential inhibitors of nsp16, a docked pose of **Z2968942047**, a derivative of pyrrolopyrimidine, is shown on Figure 10. It forms three hydrogen bonds: (1) a crucial H-bond between the acceptor of the pyrimidine ring and Cys115NH observed for SAM and sinefungin bound to nsp16, (2) an H-bond between the pyrrolic NH donor and carboxyl group of Asp114, and (3) an H-bond between the protonated pyridine ring and the carboxyl group of Asp99. **Z2968942047** also has pi-stacking between the pyrrolopyrimidine ring and phenyl ring of Phe149. The protonated positively charged pyridine of **Z2968942047** interacts with the negatively charged carboxyl group of Asp99. A distal subpocket, where the alpha-amino carboxyl fragment of sinefungin is bound, is not occupied in the case of **Z2968942047,** revealing possible sites for potency optimization.

The docked pose of **Z2045761676**, a compound with the best value of SOL scoring function, is shown in Figure 11. As in the case of **Z2968942047,** it contains two heteroaromatic rings linked by an aliphatic piperidine ring. **Z2045761676** forms four interactions: three H-bonds and pi-stacking. A pyrazole ring provides a bidentate H-bond with Cys-115NH and a carboxyl group of Asp-114. Another H-bond is found between a carboxyl group of Asp-99 and the protonated nitrogen of a pyridine ring of **Z2045761676**. Phe-149 and a pyrazole ring form pi-stacking.

## 3. Materials and Methods

### 3.1. Preparation of Ligands

Compounds from three databases were used for virtual screening: the Enamine Coronavirus Library, the Enamine Nucleoside Mimetics Library, and the Chemdiv Nucleoside Analogue Library.

The Enamine Coronavirus Library [29] contains 16,800 ligands. These ligands demonstrated good results in docking against coronavirus target proteins. We eliminated PAINS molecules and molecules containing chemical warheads. A total of 16,561 compounds were selected, and of these, 34,405 conformers were deployed in three dimensions for docking. The LigPrep module [30] was used to protonate these molecules at pH = 7.4 and generate their low-energy 3D conformers. The selection of the best ligands for further quantum-chemical processing was made in two steps. The first selection criterion, the SOL score was more negative than −6.0 kcal/mol, and the population of the first cluster (see Section 3.2) could be any number. This means that for ligands with sufficiently negative SOL scores, we did not take into account the reliability of finding the global minimum corresponding to the best ligand position. The second selection criterion, the SOL score was more negative than −5.0 kcal/mol, and the population of the first cluster (see Section 3.2) was greater than or equal to 10. This means that for ligands with sufficiently negative SOL scores, we do not take into account the population of the first cluster, which characterizes the reliability of finding the global energy minimum corresponding to the best position of the ligand. This helped prioritize only compounds with reliable docking results, and allowed not performing the second stage of screening for all compounds with scores in a range from −6.0 to −5.0 kcal/mol. Using these criteria, the 242 best ligands were selected for quantum-chemical calculations of the protein–ligand binding enthalpy. Finally, the 17 best ligands were selected for further experimental testing. They had the most negative SOL scores, the most negative binding enthalpies, and good complementarity to the active site, where the SAM cofactor bound to nsp16.

The Enamine Nucleoside Mimetics Library [31] contains 290 ligands. All these ligands are nucleoside analogues, and they can bind to polymerase or methyltransferase, as does SAM. We used 276 of these 290 ligands for docking, after filtering the reactive species, and their 2D structures were transformed into 898 3D conformers using the LigPrep module. However, none of these ligands satisfied the two criteria mentioned above, and were, therefore, not selected for experimental testing.

Third, the Chemdiv Nucleoside Analogue Library [32] contains 2600 ligands. All these ligands are nucleoside analogues. As for the other two libraries, we performed filtration of molecules with reactive groups, and for docking used 2107 of these 2600 ligands, and their 2D structures were transformed into 4047 3D conformers using the LigPrep module. Using the two criteria mentioned above, the 22 best ligands were selected for quantum-chemical calculations. The four best ligands were selected for further experimental testing. A total of 21 ligands were selected from three libraries for experimental testing.

### 3.2. Ligand Docking

In this study, docking was carried out using the SOL program [21,22]. SOL uses the preliminary calculated grid of interaction potentials between ligand and protein atoms, the MMFF94 force field, a simplified form of the generalized Born solvent model, a rigid protein approximation, and the genetic algorithm (GA) of global energy optimization. The grid of potentials is calculated in the auxiliary SOLGRID program with the size of the grid, by default, being a cube with the edge 22 Å covering the active site of the protein. The flexibility of a ligand includes translation and rotation of a ligand as a rigid body and rotations of each torsion. Conformations of macrocycles are not sampled. The target function in solving the global optimization problem is the sum of the energy of protein–ligand interaction and the ligand internal strain energy. The main parameters of GA are the following: population size is 30,000 by default, the number of generations is 1000, and the number of independent runs of GA is 50 by default. The latter is needed to estimate the reliability of the global minimum finding, as follows. After docking a ligand, 50 solutions of the global optimization problem are obtained. These 50 solutions to the global optimization problem correspond to 50 ligand positions in the active site of the protein, corresponding to the lowest values of the target energy function that are found in each GA run. These ligand positions are clustered by RMSD <1 Å for each pair of ligand positions, and the clusters are ranked by the lowest energy corresponding to ligand positions in each cluster. Therefore, the ligand position corresponding to the global energy minimum is in the first cluster. Obviously, the reliability of the global energy minimum finding is high if the population of the first cluster is relatively large. This means that in several independent GA runs, practically the same ligand position is found corresponding to the lowest energy value. In the limited case of fifty different clusters, one cannot assume that a global minimum has been found. The same can be said in the case of a smaller number of clusters, when the population of the first cluster is 1. In the latter case, docking can be repeated with higher GA parameters and with an increased number of GA runs. If the same position of the ligand corresponding to the global minimum is obtained, and the population of the first cluster is greater than 1, then the found global minimum can be considered reliable, and docking is successful. The scoring function of SOL is a weighted sum of physics-based terms (van-der-Waals, electrostatic, desolvation) and a simplified entropy term (a number of torsions), with coefficients calculated in such way to best reproduce known Gibbs binding free energies from a test dataset of protein–ligand complexes.

The SOL program was adapted for virtual screening on the Lomonosov-2 supercomputer of the Lomonosov Moscow State University [26]. Depending on the size and number of ligand torsions, the docking of one ligand on one computational core takes between one and several hours. The parallel version of SOL, based on MPI (message passing interface), allows docking of one ligand using many hundreds of supercomputer cores. This can be useful for docking with increased GA parameters. For virtual screening of large databases of ligands, a sequential version of the SOL program is used: each ligand is docked on a separate computing core.

### 3.3. Protein–Ligand Binding Enthalpy

The enthalpy of protein–ligand binding was calculated using the MOPAC program [33], using the PM7 semi-empirical quantum chemical method [34], and taking into account the interaction with an aqueous solvent in the COSMO model [35]. The binding enthalpy $\Delta H_{bind}$ was calculated as follows:(1)$$\Delta H_{bind}=\Delta H_{protein-ligand}-\left(\Delta H_{protein}+\Delta H_{ligand}\right),$$

$\Delta H_{protein-ligand}$ is the enthalpy of formation of the protein–ligand complex; $\Delta H_{ligand}$ is the enthalpy of formation of the unbound ligand; $\Delta H_{protein}$ is the enthalpy of formation of unbound protein. In this equation, $\Delta H_{protein}$ is calculated for the same conformation of the protein that was used for docking, $\Delta H_{ligand}$ is calculated for the unbound ligand conformation with the lowest PM7+COSMO energy, and $\Delta H_{protein-ligand}$ is calculated as follows. The local optimization of the energy of the protein–ligand complex using the PM7 method is carried out from the ligand pose that has been found during docking, when the positions of all ligand atoms are varied. In the identified local minimum, the energy of the complex was recalculated using PM7 and the COSMO model.

The total computation time spent on screening was about 149,000 CPU × hours, including SOL and MOPAC calculations.

## 4. Conclusions

In this work, an atomistic model of nsp16 SARS-CoV-2 was constructed and verified using native docking and cross-docking. This model was used in the search for new low-molecular-weight inhibitors of nsp16 in commercially available databases, using docking followed by quantum-chemical calculation of the enthalpy of protein–ligand binding. The best 21 candidates to become inhibitors of nsp16 were selected for experimental testing. These inhibitors belong to several chemical classes. Some of them are nucleoside analogs: indazole (**Z2003934806**, **Z2394358664**) and pyrrolopyrimidine (**Z2968942047**). Three compounds **8012-4548**, **C692-0494**, and **E714-0026** are purines, and the other compounds belong to various chemical classes. All of the selected compounds have not yet been published as inhibitors of SARS-CoV-2 nsp16. The findings of the study can accelerate the design of antiviral agents against SARS-CoV-2.

After confirmation of activity, the next iterations will include structural optimization of the potency for validated hits. The protein model prepared at this stage can accelerate optimization by revealing the places in an active site that can be exploited to favor binding. Design of SAM-competitive inhibitors of nsp16 should also be accompanied by the crucial step of a selectivity check for human methyltransferases, since they bind to the same cofactor. From this perspective, protein models of human methyltransferases can be prepared for docking in the next steps of the design.

## Figures and Tables

**Figure 1 molecules-27-02721-f001:**
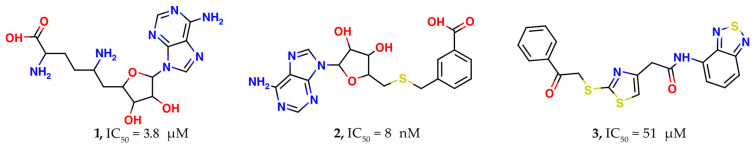
Experimentally confirmed nsp16 inhibitors.

**Figure 2 molecules-27-02721-f002:**
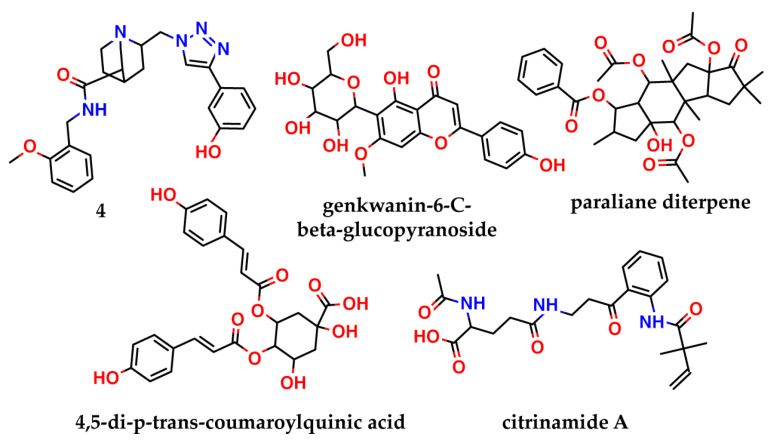
Inhibitors of nsp16 predicted by molecular modeling in the literature, with no confirmation of activity.

**Figure 3 molecules-27-02721-f003:**
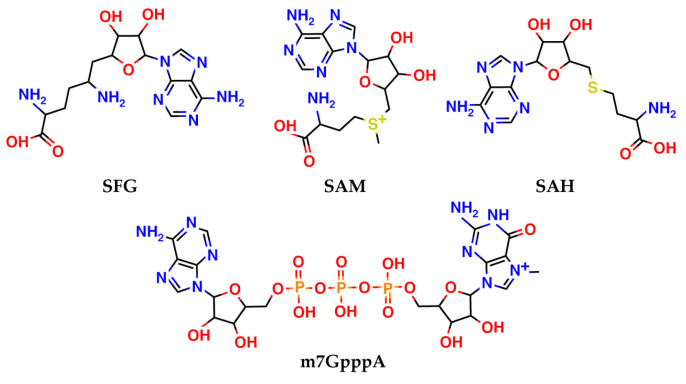
Ligands crystallized together with nsp16 in the Protein Data Bank.

**Figure 4 molecules-27-02721-f004:**
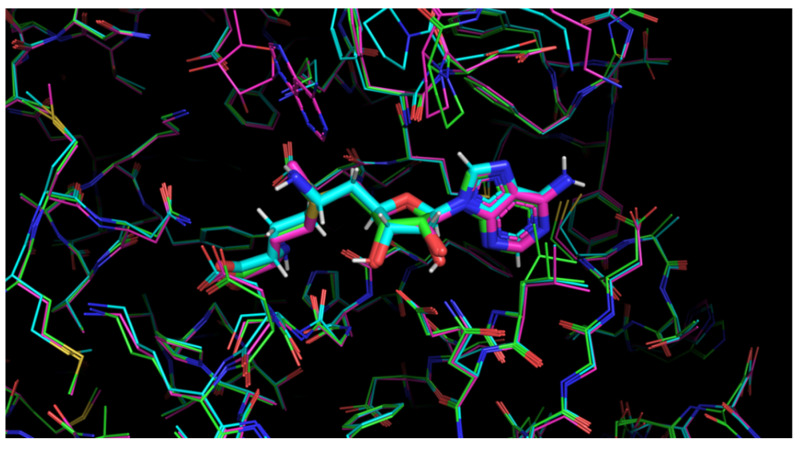
Alignment of the three complexes: 6WKQ (green), 6YZ1 (blue), and 6WKS (pink). The figure was made in PyMOL (version 2.5.0a0).

**Figure 5 molecules-27-02721-f005:**
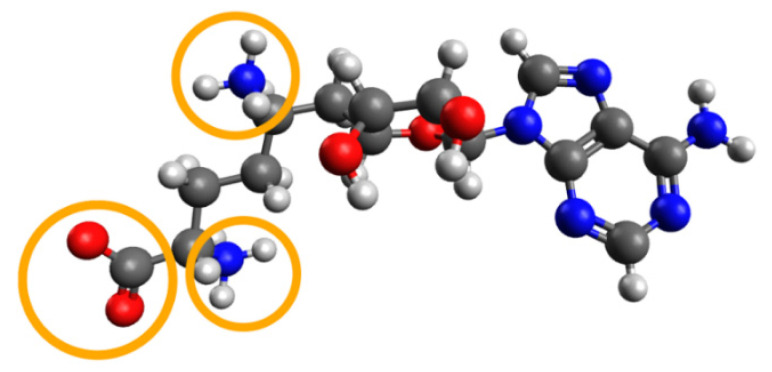
Protonated form of sinefungin at pH = 7.4. Charged groups are highlighted by orange circles. Protonation was carried out with the molecule editor and visualizer Avogadro [23].

**Figure 6 molecules-27-02721-f006:**
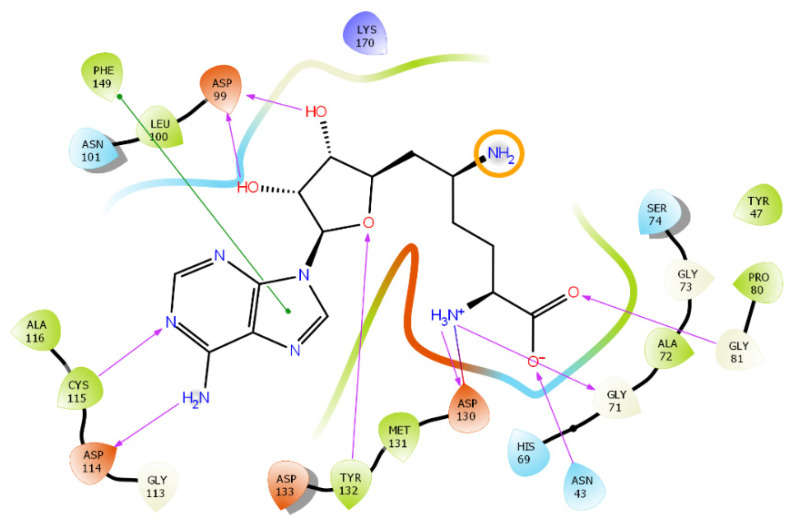
Interaction profile of sinefungin with nsp16. A charged amino group that is in a solvent-accessible area and does not interact with the protein is highlighted by an orange circle. Pink lines indicate H-bonds; a green line signifies pi-stacking. Figure was prepared in Maestro [24].

**Figure 7 molecules-27-02721-f007:**
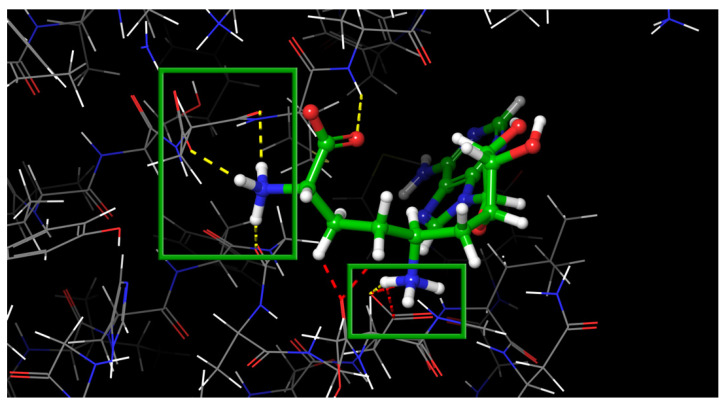
Fully charged native ligand from 6YZ1 after docking. Ionic interactions with charged groups are fully saturated and highlighted by green boxes. The figure was made in PyMOL (version 2.5.0a0).

**Figure 8 molecules-27-02721-f008:**
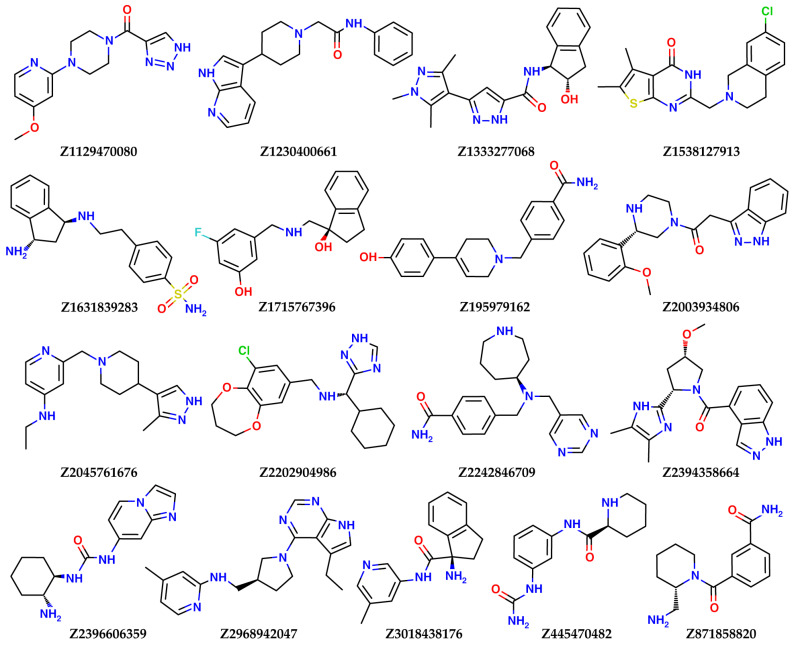
The best compounds selected from the Enamine Coronavirus Library for experimental testing on nsp16 inhibitory activity.

**Figure 9 molecules-27-02721-f009:**
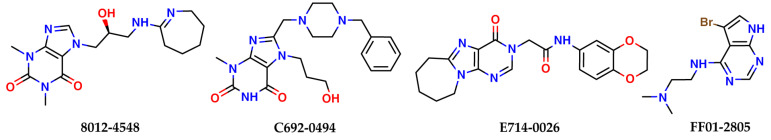
The best compounds selected from the Chemdiv Nucleoside Analogue Library for experimental testing of nsp16 inhibitory activity.

**Figure 10 molecules-27-02721-f010:**
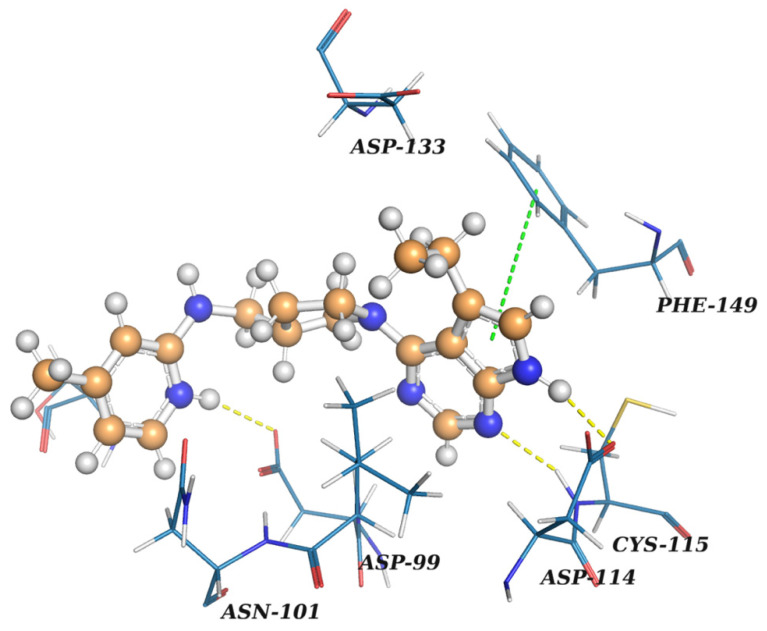
A docked pose of **Z2968942047** in the SAM-binding site of nsp16. The ligand is shown as a ‘ball-and-stick’ model and colored in brown. The protein residues are represented as lines. Yellow dashed lines indicate H-bonds. A green line signifies pi-stacking. The figure was prepared in PyMol [28].

**Figure 11 molecules-27-02721-f011:**
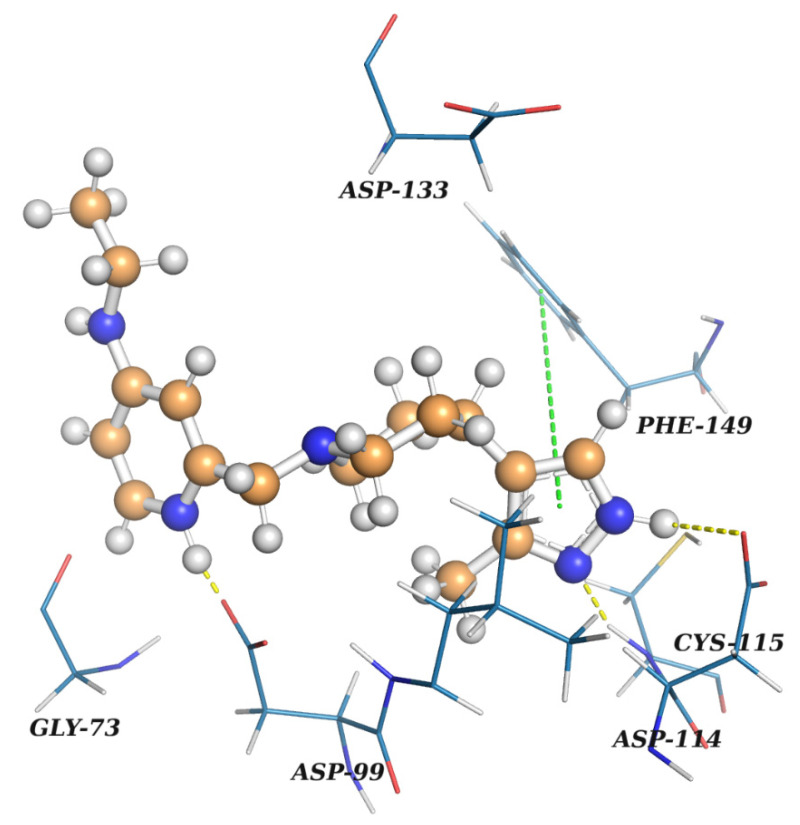
A docked pose of **Z2045761676** in the SAM-binding site of nsp16. The ligand is shown as a ‘ball-and-stick’ model and colored in brown. The protein residues are represented as lines. Yellow dashed lines indicate H-bonds. A green line signifies pi-stacking. The figure was prepared in PyMol.

**Table 1 molecules-27-02721-t001:** Deposited complexes of SARS-CoV-2 nsp16 in PDB.

PDB ID	Ligand	Resolution, Å
6W4H	SAM	1.8
6W61	SAM	2
6W75	SAM	1.95
7C2J	SAM	2.8
6WKQ	SFG	1.98
6YZ1	SFG	2.4
6WKS	SAM + RNA substrate	1.8
6WJT	SAH	2

**Table 2 molecules-27-02721-t002:** Features of the crystal complexes chosen for preparation of the fully-atomic nsp16 model.

PDB ID	Resolution, Å	R-Value Work	Missing Elements
6YZ1	2.4	0.187	One residue on N- and C-terminus
6WKQ	1.98	0.162	Four residues missed on N-terminus

**Table 3 molecules-27-02721-t003:** Native docking results for a model based on 6YZ1 and 6WKQ.

Protein	Ligand	RMSD, Å	1st Cluster Popul.	N. of Clusters	Score, kcal/mol
6YZ1	SFG (fully charged)	4.02	2	45	−4.89
6WKQ	SFG (fully charged)	10.28	2	42	−4.66

**Table 4 molecules-27-02721-t004:** Native docking results for a model based on 6YZ1 and 6WKQ, with partially neutralized SFG.

Protein	Ligand	RMSD, Å	1st Cluster Popul.	N. of Clusters	Score, kcal/mol
6YZ1	SFG (neutralized)	4.0197	2	45	−4.89
6WKQ	SFG (neutralized)	0.87	1	38	−7.05
6WKQ*	SFG (neutralized)	0.87	8	36	−7.07

**Table 5 molecules-27-02721-t005:** Cross-docking of SAH into the model constructed on the basis of the 6WKQ complex: docking with the default GA parameters—6WKQ, docking with the elevated GA parameters—6WKQ*.

Protein	Ligand	RMSD, Å	1st Cluster Popul.	N. of Clusters	Score, kcal/mol
6WKQ	SAH (quasi-native—from 6WJT complex)	2.54	1	44	−5.63
6WKQ*	SAH (quasi-native—from 6WJT complex)	0.97	9	57	−7.48

**Table 6 molecules-27-02721-t006:** Results of calculations for the best compounds selected for experimental testing for nsp16 inhibitory activity.

Ligname	Score, kcal/mol	1st Cluster Popul.	ΔH Bind, kcal/mol
**Reference (SFG)**	−7.07	8	−93.8
**Enamine Coronavirus Library**			
**Z1129470080**	−6.04	36	−38.3
**Z1230400661**	−5.26	20	−51.2
**Z1333277068**	−5.24	47	−53.8
**Z1538127913**	−6.13	34	−40.7
**Z1631839283**	−6.38	2	−62.5
**Z1715767396**	−5.38	23	−53.0
**Z195979162**	−5.24	40	−53.4
**Z2003934806**	−6.41	30	−44.8
**Z2045761676**	−6.76	4	−30.8
**Z2202904986**	−5.94	17	−60.8
**Z2242846709**	−5.51	30	−61.6
**Z2394358664**	−6.11	32	−51.6
**Z2396606359**	−5.81	30	−54.5
**Z2968942047**	−6.15	17	−55.4
**Z3018438176**	−6.43	12	−42.3
**Z445470482**	−5.54	15	−60.0
**Z871858820**	−5.70	38	−54.7
**Chemdiv Nucleoside Analogue Library**			
**8012-4548**	−5.41	33	−39.3
**C692-0494**	−5.27	10	−48.4
**E714-0026**	−5.00	10	−43.2
**FF01-2805**	−5.03	37	−37.7

## Data Availability

Not applicable.

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
