# Peer review of "Novel Inhibitors of 2′-O-Methyltransferase of the SARS-CoV-2 Coronavirus"

_molecules, 2022, doi:10.3390/molecules27092721_

Round 1

Reviewer 1 Report

The results of virtual screening of ca. 40,000 molecules from Enamine and ChemDiv libraries are presented. Supercomputer highthroughput docking of potential coronavirus 2′O-methyltransferase (nsp16) inhibitors was carried out with the specialized software SOL.

In addition to the "classic docking" based on molecular mechanics, ligand-target binding enthalpy was estimated using semiempirical quantum chemical calculations.

Twenty-one hits with reasonable scoring function values were selected for experimental testing of their inhibiting activity.

In general, the study corresponds to the up-to-the-date level of in silico virtual screening. However, some minor improvement of the results’ presentations still needed.

  1. It is necessary to add more detailed description of the SOL software because it may be not well-known for the readers.
  2. It is necessary to add in the Table 6 the scoring function values for the reference ligand (if any), or explain why such addition is not relevant in the particular case.

Certainly, all results of in silico screening must be validated in the 2′O-methyltransferase inhibiting assays and in viral replication inhibiting assay; however, I assume that this study will be carried out in the near future.

Reviewer 2 Report

The manuscript of Sulimov et al. reports in silico study of novel inhibitors of 2’-O-methyltransferase enzyme of SARS-CoV-2. Despite particular advances and treatment options, it is still a hot topic that deserves to be investigated.

The manuscript is generally well-written and concise. In my opinion, it may be accepted for publication in Molecules journal (special issue Recent Trends on Enzymes Inhibitors and Activators in Drug Research 3.0) after careful addressing my following comments and notes:

  • in my opinion, it is correct to write the name of the title enzyme is 2’-O-methyltransferase
  • please, add the Enzyme Commission number of it,
  • some abbreviations are not properly defined, e.g., SARS, COVID, ATP,
  • some nomenclature descriptors should be italicized, e.g., S-adenosyl, 2’-O-methyltransferase, N-methyl, p-, trans-, 6-C,
  • there are some typos, e.g., SARS-COV-2 (line 37), IC50 (line 67), missing spaces (lines 233 and 244). Please, check the manuscript carefully to avoid these.
  • Figure 1 – please, add IC50 value for the inhibitor 1,
  • please, provide structures of triazine inhibitor (mentioned on line 89) and four inhibitors mentioned on lines 101-103,
  • INN names of drugs should not be capitalized (line 94),
  • Nsp16 is not written uniformly, somewhere it is as NSP16,
  • missing more detailed specification of Avogadro program (line 154),
  • 6 – missing source program,
  • line 244 – please, provide reference(s) that azaindoles are typical nucleoside analogues used in drug design,
  • there are many mistakes in the References section – missing pages/article numbers, volumes, use of bold font, incorrect abbreviations, (e.g., refs 4, 10, 14, 16, 20, 26, 36, 38, etc.

Moreover, I found inappropriate self-citations by authors. Ten references from a total of 41 are co-authored by Sulimov. From this point of view, the sentence „The SOL program has been successfully used to develop inhibitors of thrombin [32], urokinase (uPA) [33,34], and coagulation factors Xa, XIa and XIIa [35–38].“ is in my opinion inadequate and redundant self-citations should be omitted.

In addition to these comments, please add a specific short paragraph on your future perspectives and application of this research.

Reviewer 3 Report

In this paper, the authors have searched for molecules able to inhibiting SARS-CoV-2 by interacting with the SAM binding site of the nsp16 2’O-methyltransferase of the virus. For this purpose, they have conducted virtual screening experiments with chemical compounds from three chemical libraries on a crystallographic structure of nsp16. By using the docking score and an evaluation of the enthalpy of protein-ligand binding with semiempirical quantum calculation, they proposed 21 compounds as potential nsp 16 inhibitors.

What is quite disappointing for the reviewer is that, although the introduction was well documented, describing precisely other similar studies in literature  and clearly highlighting the problem of the non-specificity of inhibitors identified previously, none answer to this problematic is suggested by the actual work, that in fact seems to only propose new potential nsp16 binders that moreover need to be experimentally validated. It is thus hard to understand what is the gain bring by this new in silico study of nsp16 potential inhibitors.

The work is certainly well conducted, with a careful preparation of the protein target and of the chemical libraries and a cautious attention is devoted to the docking protocol validation with an assessment of the docking score reinforced by semiempirical calculations. 

However, several points seem showing that the docking software used in this paper is not really well adapted to the studied system. First, as said by the sentence l. 128, there is no defined parameters for the SAM molecule in the SOL docking program, preventing the use of 5 over the 8 available nsp16-ligand structures. Second, the authors show p. 6, 7 that the redocking of the two chosen systems failed when using the default parameters of the SOL docking software, needing the use of elevated parameters to obtain better results. This conducts to a drastic increase of the computational time, as said l. 204, of about 5hours using 512 cores for the redocking of SFG in 6WKQ and, l. 222, of about 3h43 for the cross-docking of SAH in 6WKQ. Therefore, I wonder about the total computational time needed to perform virtual screening of the selected chemical libraries. Does it not exist docking software more adapted to treat such systems?

Another point that is a bit confuse to me concerns the two-step  protocol used for virtual screening. The authors described l. 282 to 288:

  • The first selection criterion, the SOL score was more negative than −6.0 kcal/mol, and the population of the first cluster could be any.
  • The second selection criterion, the SOL score was more negative than −5.0 kcal/mol, and the population of the first cluster was greater than or equal to 10.

I suppose that this two criterion are applied on two successive runs of docking, am I right? If it is the case, does it mean that the first run of docking use GA default parameters and the second GA elevated parameters? If not, which parameters are used? It remains unclear for me.

I also find regrettable that very few analysis were presented about the 21 selected ligands. Indeed, the interactions between the docked pose and the protein are only shown for one ligand and no comparison is possible with the interaction profile of sinefungin as the nsp16 residues are not labeled identically on figures 5 and 9. I hypothesise that Phe6947 (fig 5) correspond to Phe149 (fig 9), Asp6912 to Asp114, Asp6897 to Asp99 etc… But the authors have to harmonise their labels between the figures. I have to notice that they mentioned a crucial H-bond between sinefungin and Cys115 which however does not appear Fig 5 (with corresponding Cys6913 I think). Moreover, if the idea is to design a more select and potent inhibitor than sinefungin, I am not convinced by the presented compound which occupies a smaller volume than the reference ligand. In that regard, I remarked that the authors claims they used compounds with « low molecular weight » for the virtual screening (l. 106, 229 and 363) but they never provide information about the meaning of this term and about the reasons of this choice.

In conclusion, I find this work rigorous and well conducted but I can hardly understand the gain of this study for the scientific community, compared to the previous studies of nsp16 inhibitors, without any experimental validation of activity nor information about the selectivity of the new proposed compounds.

Reviewer 4 Report

The authors present an interesting and attractive manuscript for designing new compounds.
The text is understandable, although I would like they could improve the discussion of the results from a more structural point of view for the most active compounds.
I find "curios" the problem of protonation of crystallized ligands and the neutralized state is the most suitable for simulation.
I would have loved to access the SOL program to repeat the most interesting results.

Round 2

Reviewer 2 Report

The authors have accepted and taken into account the comments of the opponent, so the manuscript can be accepted for publication in the journal Molecules.

Reviewer 3 Report

The authors have improved their manuscript and clarify the points that seemed doubtful for the reviewer. Nevertheless, although the authors claims that the interest of this study is justify by the sentence "All selected compounds have not yet been published previously as inhibitors of SARS-CoV-2 nsp16 », it seems insufficient in regard to the fact that no experiment supports the calculations. It is certain that the discover of new compounds inhibiting nsp16 will be of great interest, provided that they are active.  Unfortunately, docking methods may failed in such prediction and thus confirmation with experimental data is a non-neglectable added value. Such rigorous and well-conducted study is very encourageous and it is regrettable not having the final result.

I think that the present work may be published, provided the authors insist in their conclusion on the fact that such proposal compounds are potential lead that need absolutely experimental confirmation before being considered as real leads and also before considering this model as a way to accelerate design of antiviral agents against SARS-CoV-2.